# Biobank-wide association scan identifies risk factors for late-onset Alzheimer's disease and endophenotypes

**Donghui Yan[1†], Bowen Hu[2†], Burcu F Darst[3], Shubhabrata Mukherjee[4], Brian W Kunkle[5], Yuetiva Deming[3], Logan Dumitrescu[6], Yunling Wang[1], Adam Naj[7], Amanda Kuzma[7], Yi Zhao[7], Hyunseung Kang[2], Sterling C Johnson[8,9,10], Cruchaga Carlos[11], Timothy J Hohman[6], Paul K Crane[4], Corinne D Engelman[3,8,10], Alzheimer's Disease Genetics Consortium (ADGC), Qiongshi Lu[2,12]\***

[1]University of Wisconsin-Madison, Madison, United States; [2]Department of Statistics, University of Wisconsin-Madison, Madison, United States; [3]Department of Population Health Sciences, University of Wisconsin-Madison, Madison, United States; [4]Division of General Internal Medicine, Department of Medicine, University of Washington, Seattle, United States; [5]University of Miami Miller School of Medicine, Miami, United States; [6]Vanderbilt Memory and Alzheimer's Center, Vanderbilt University Medical Center, Vanderbilt University School of Medicine, Nashville, United States; [7]School of Medicine, University of Pennsylvania, Philadelphia, United States; [8]Wisconsin Alzheimer's Institute, University of Wisconsin School of Medicine and Public Health, Madison, United States; [9]Geriatric Research Education and Clinical Center, Wm. S. Middleton Memorial VA Hospital, Madison, United States; [10]Alzheimer's Disease Research Center, University of Wisconsin School of Medicine and Public Health, Madison, United States; [11]Department of Psychiatry, Washington University in St. Louis, St. Louis, United States; [12]Department of Biostatistics and Medical Informatics, University of Wisconsin-Madison, Madison, United States

**\*For correspondence:**
qlu@biostat.wisc.edu

[†]These authors contributed equally to this work

**Competing interest:** The authors declare that no competing interests exist.

**Abstract** Rich data from large biobanks, coupled with increasingly accessible association statistics from genome-wide association studies (GWAS), provide great opportunities to dissect the complex relationships among human traits and diseases. We introduce BADGERS, a powerful method to perform polygenic score-based biobank-wide association scans. Compared to traditional approaches, BADGERS uses GWAS summary statistics as input and does not require multiple traits to be measured in the same cohort. We applied BADGERS to two independent datasets for late-onset Alzheimer's disease (AD; n=61,212). Among 1738 traits in the UK biobank, we identified 48 significant associations for AD. Family history, high cholesterol, and numerous traits related to intelligence and education showed strong and independent associations with AD. Furthermore, we identified 41 significant associations for a variety of AD endophenotypes. While family history and high cholesterol were strongly associated with AD subgroups and pathologies, only intelligence and education-related traits predicted pre-clinical cognitive phenotypes. These results provide novel insights into the distinct biological processes underlying various risk factors for AD.

## eLife assessment

In the last 15 years, large-scale association studies (GWAS) have served to estimate the association between genome-wide common variants and a large number of disparate traits and diseases in humans. This **valuable** method provides a new way to find correlations between the genetic

component of a phenotype of interest, and all this wealth of genetic information. This software adds as a new tool to investigate genetic correlation between traits, and to generate new mechanistic hypotheses and dissect the role of the observed associations in disease heterogeneity. The results of the application of their method are **solid** and generally agree with what others have seen using similar AD and UKB data.

## Introduction

Late-onset AD is a prevalent, complex, and devastating neurodegenerative disease without a current cure. Millions of people are currently living with AD worldwide, and the number is expected to grow rapidly as the population continues to age (*Prince et al., 2013*; *Reitz and Mayeux, 2014*). With the failure of numerous drug trials, it is of great interest to identify modifiable risk factors that can be potential targets in the therapeutics development for AD (*Østergaard et al., 2015*; *Larsson et al., 2017*; *Norton et al., 2014*). Epidemiological studies that directly test associations between measured risk factors and AD are difficult to conduct and interpret because identified associations are, in many cases, affected by confounding and reverse causality. Despite being ubiquitous challenges in risk factor studies for complex diseases, these issues are particularly critical for AD due to its extended pre-clinical stage – irreversible pathologic changes have already occurred in the decade or two prior to clinical symptoms (*Jack et al., 2013*). On the other hand, Mendelian randomization methods (*Sleiman and Grant, 2010*; *Davey Smith and Hemani, 2014*; *Zhu et al., 2018*) have been developed to identify causal risk factors for disease using data from GWAS. Despite the favorable theoretical properties in identifying causal relationships, these methods have limited statistical power, thereby not suitable for hypothesis-free screening of risk factors.

Motivated by transcriptome-wide association study – an analysis strategy that identifies genes whose genetically regulated expression values are associated with disease (*Gamazon et al., 2015*; *Gusev et al., 2016*; *Hu et al., 2018*), we seek a systematic and statistically powerful approach to identify risk factors using summary association statistics from large-scale GWAS. GWAS for late-onset AD has been successful, and dozens of associated loci have been identified to date (*Lambert et al., 2013*; *Harold et al., 2009*; *Hollingworth et al., 2011*; *Naj et al., 2011*; *Seshadri et al., 2010*; *Jun et al., 2017*). Although direct information on risk factors is limited in these studies, dense genotype data on a large number of samples, in conjunction with independent reference datasets for thousands of complex human traits such as the UK biobank (*Bycroft et al., 2017*), make it possible to genetically impute potential risk factors and test their associations with AD. This strategy allows researchers to study risk factors that are not directly measured in AD studies. Furthermore, it reduces the reverse causality because the imputation models are trained on independent, younger, and mostly dementia-free reference cohorts, thereby improving the interpretability of findings.

Here, we introduce BADGERS (Biobank-wide Association Discovery using GEnetic Risk Scores), a statistically powerful and computationally efficient method to identify associations between a disease of interest and a large number of genetically imputed complex traits using GWAS summary statistics. We applied BADGERS to identify associated risk factors for AD from 1738 heritable traits in the UK biobank and replicated our findings in independent samples. Furthermore, we performed multivariate conditional analysis, Mendelian randomization, and follow-up association analysis with a variety of AD biomarkers, pathologies, and pre-clinical cognitive phenotypes to provide mechanistic insights into our findings.

## Results
### Method overview

Here, we briefly introduce the BADGERS model. The workflow of BADGERS is shown in *Figure 1*. A brief flowchart including all the analyses we contained in the manuscript was shown in the supplementary material (*Figure 1—figure supplement 1*). Complete statistical details are discussed in the **Methods** section. BADGERS is a two-stage method to test associations between traits. First, polygenic risk scores (PRS) are trained to impute complex traits using genetic data. Next, we test the association between a disease or trait of interest and various genetically-imputed traits. Given a PRS model, the imputed trait can be denoted as

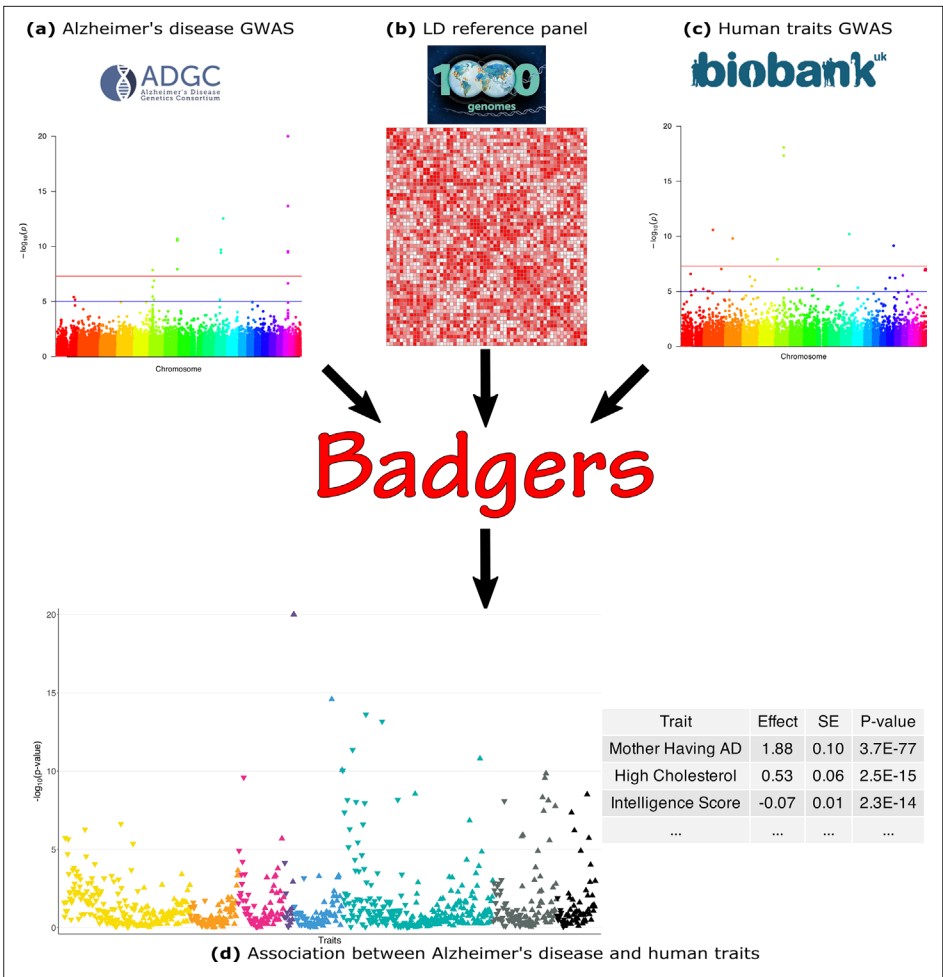

**Figure 1.** Biobank-wide Association Discovery using GEnetic Risk Scores (BADGERS) Workflow. BADGERS takes (**a**) Alzheimer's disease genome-wide association studies (GWAS), (**b**) linkage disequilibrium (LD) reference panel, and (**c**) Human traits GWAS from the UK biobank as input. The generated result will be the (**d**) Association between Alzheimer's disease and human traits. In graph (**d**), each triangle represents one human trait, and different colors represent different trait categories.

The online version of this article includes the following figure supplement(s) for figure 1:

**Figure supplement 1.** A flowchart for analyses of Alzheimer's genetic data.

$$\hat{T} = XW$$

where $X_{N \times M}$ is the genotype matrix for $N$ individuals in a GWAS, and $W_{M \times 1}$ is the Mx1 matrix denotes pre-calculated weight values on SNPs in the PRS model. Then, we test the association between measured trait $Y$ and imputed trait $T$ via a univariate linear model.

$$Y = \alpha + \hat{T}\gamma + \delta$$

The test statistic for $\gamma$ can be expressed as:

$$Z = \frac{\hat{\gamma}}{se\left(\hat{\gamma}\right)} \approx W^T \Gamma \widetilde{Z}$$

where $\widetilde{Z}$ is the vector of SNP-level association z-scores for trait $Y$, and $\Gamma$ is a diagonal matrix with the j$^{th}$ diagonal element being the ratio between the standard deviation of the j$^{th}$ SNP and that of $\hat{T}$ .

This model can be further generalized to perform multivariate analysis. If $K$ imputed traits are included in the analysis, we use a similar notation as in univariate analysis:

$$\hat{T}^* = XW^*$$

Here, $W_{M \times K}^*$ is a matrix and each column of $W^*$ is the pre-calculated weight values on SNPs for each imputed trait. Then, the associations between $Y$ and $K$ imputed traits $\hat{T}_i$ $(1 \leq i \leq K)$ are tested via a multivariate linear model.

$$Y = \alpha^* + \hat{T}^* \gamma^* + \delta^*$$

where $\gamma^* = (\gamma_1, \ldots, \gamma_K)^T$ is the vector of regression coefficients. The z-score for $\gamma_i$ $(1 \leq i \leq K)$ can be denoted as:

$$Z_i = \frac{\hat{\gamma}_i}{se(\hat{\gamma}_i)} \approx \frac{1}{\sqrt{U_{ii}}} I_i^T U (W^*)^T \Theta \widetilde{Z}$$

where $U$ is the inverse variance-covariance matrix of $\hat{T}^*$ ; $I_i$ is the $K \times 1$ vector with the $i^{th}$ element being 1 and all other elements equal to 0; is a $M \times M$ diagonal matrix with the $i^{th}$ diagonal element being $\sqrt{var(X_i)}$ ; and $\widetilde{Z}$ is defined the same as the univariate case as the vector of SNP-level association z-scores for trait $Y$.

## Simulations

We used real genotype data from the Genetic Epidemiology Research on Adult Health and Aging (GERA) to conduct simulation analyses (**Methods**). First, we evaluated the performance of our method

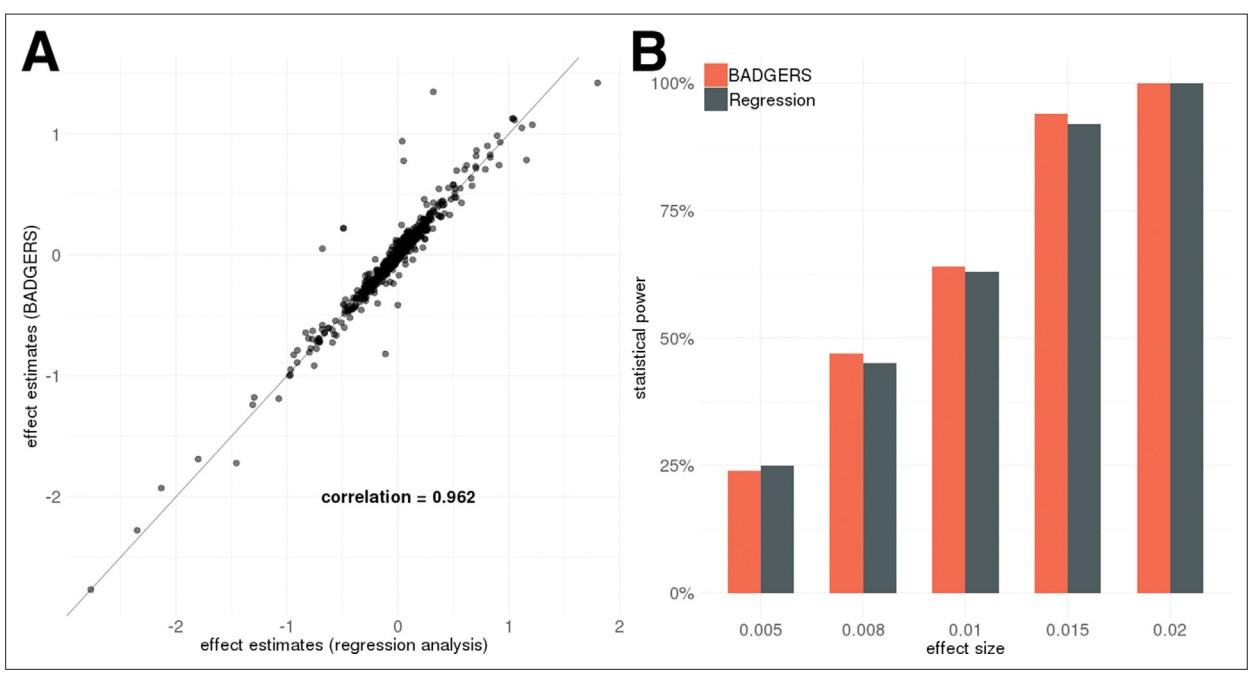

**Figure 2.** Simulation results. Biobank-wide Association Discovery using GEnetic Risk Scores (BADGERS) and regression analysis based on individual-level data showed (**A**) highly consistent effect size estimates for 1738 polygenic risk scores (PRS) in simulation and (**B**) comparable statistical power (setting 3).

The online version of this article includes the following figure supplement(s) for figure 2:

**Figure supplement 1.** Comparison of effect size estimates from Biobank-wide Association Discovery using GEnetic Risk Scores (BADGERS) and regression analysis based on individual-level data.

**Figure supplement 2.** Comparison of p-values from Biobank-wide Association Discovery using GEnetic Risk Scores (BADGERS) and regression analysis based on individual-level data.

**Figure supplement 3.** Comparison of effect size estimates from Biobank-wide Association Discovery using GEnetic Risk Scores (BADGERS) and regression analysis based on individual-level data when p-values are smaller than 0.05.

**Figure supplement 4.** Biobank-wide Association Discovery using GEnetic Risk Scores (BADGERS) estimates using marginal polygenic risk scores (PRS) and joint PRS.

on data simulated under the null hypothesis. We tested the associations between randomly simulated traits and 1738 PRS from the UK biobank and did not observe inflation in type-I error (***Supplementary file 1***). Similar results were also observed when we simulated traits that are heritable but not directly associated with any PRS. Since BADGERS only uses summary association statistics and externally estimated linkage disequilibrium (LD) as input, we also compared effect estimates in BADGERS with those of traditional regression analysis based on individual-level data. Regression coefficient estimates and association p-values from these two methods were highly consistent in both simulation settings (***Figure 2A*** and ***Figure 2—figure supplements 1–3***), showing minimal information loss in summary statistics compared to individual-level data indicating highly consistent performance compared to methods based on individual-level data. To evaluate the statistical power of BADGERS, we simulated traits by combining effects from randomly selected PRS and a polygenic background (**Methods**). We set the effect size of PRS to be 0.02, 0.015, 0.01, 0.008, and 0.005. BADGERS showed comparable statistical power to the regression analysis based on individual-level genotype and phenotype data (***Figure 2B***, ***Supplementary file 1***). Overall, our results suggest that using summary association statistics and externally estimated LD as a proxy for individual-level genotype and phenotype data does not inflate type-I error rate or decrease power. The performance of BADGERS is comparable to regression analysis based on individual-level data. We also studied if more sophisticated polygenic risk prediction methods could potentially lead to higher statistical power in downstream association tests. We compared the performance of PRS based on marginal effect sizes with that of LDpred, a more sophisticated PRS model that jointly estimates SNP effects via a Bayesian framework (***Vilhjálmsson et al., 2015***). Imputation models based on multivariate analysis indeed improved the results. When using marginal PRS to impute traits, the correlation between $\hat{\gamma}_i$ and $\gamma_i$ was 0.79. This correlation improved to 0.91 when using LDpred PRS (***Figure 2—figure supplement 4***). However, such improvement did not substantially affect the statistical power in association testing. Using marginal PRS, our analysis achieved a statistical power of 86% to identify associations at a type-I error rate of 0.05, and the power was 88% when using multivariate effect estimates to calculate PRS. These results suggest that while more sophisticated PRS methods can improve the results in BADGERS, simple PRS based on marginal effects also shows reasonably good performance.

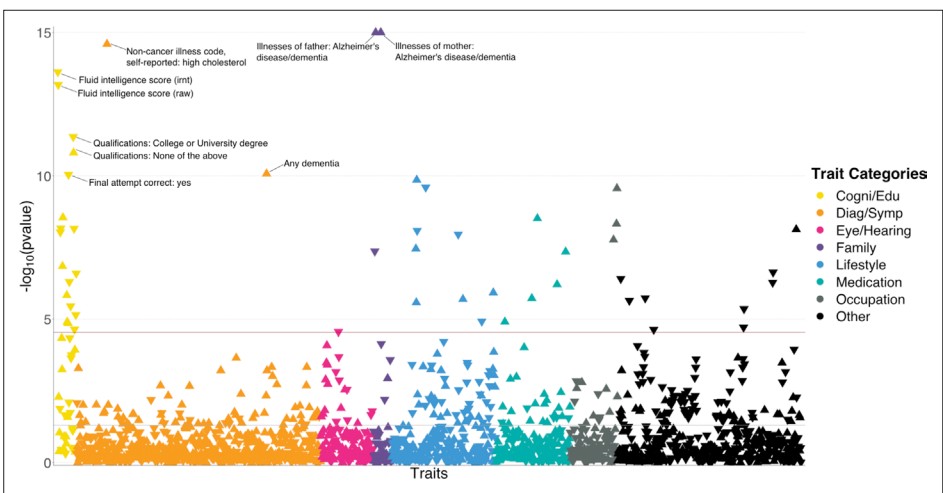

**Figure 3.** polygenic risk score (PR)S-based biobank-wide association scan (BWAS) identifies risk factors for Alzheimer's disease (AD). Meta-analysis p-values for 1738 heritable traits in the UK biobank are shown in the figure. p-values are truncated at 1e-15 for visualization purposes. The horizontal line marks the Bonferroni-corrected significance threshold (i.e. p=0.05/1738). Positive associations point upward, and negative associations point downward.

The online version of this article includes the following figure supplement(s) for figure 3:

**Figure supplement 1.** Workflow of the two-stage biobank-wide association scan (BWAS) for late-onset Alzheimer's disease (AD).

**Figure supplement 2.** Associations between Alzheimer's disease (AD) and education attainment in two independent analyses.

## Identify risk factors for late-onset AD among 1738 heritable traits in the UK biobank

We applied BADGERS to conduct a biobank-wide association scan (BWAS) for AD risk factors from 1738 heritable traits (p<0.05; **Methods**) in the UK biobank. We repeated the analysis on two independent GWAS datasets for AD and further combined the statistical evidence via meta-analysis (*Figure 3—figure supplement 1*). We used stage-I association statistics from the International Genomics of Alzheimer's Project (IGAP; n=54,162) as the discovery phase, then replicated the findings using 7050 independent samples from the Alzheimer's Disease Genetics Consortium (ADGC). We identified 50 significant trait-AD associations in the discovery phase after correcting for multiple testing, among which 14 had p<0.05 in the replication analysis. Despite the considerably smaller sample size in the replication phase, top traits identified in the discovery stage showed strong enrichment for p<0.05 in the replication analysis (enrichment = 2.5, p=2.2e-5; hypergeometric test). In the meta-analysis, a total of 48 traits reached Bonferroni-corrected statistical significance and showed consistent effect directions in the discovery and replication analyses (*Figure 3* and *Supplementary file 2*).

Unsurprisingly, many identified associations were related to dementia and cognition. Family history of AD and dementia showed the most significant associations with AD (p=3.7e-77 and 5.2e-28 for illnesses of mother and father, respectively). Having any dementia diagnosis is also strongly and positively associated (p=8.5e-11). In addition, we observed consistent and negative associations between better performance in cognition test and AD risk. These traits include fluid intelligence score (p=2.4e-14), time to complete round in cognition test (p=2.8e-9), correct final attempt (p=9.1e-11), and many others. Consistently, education attainment showed strong associations with AD. Age completed full time education (p=2.5e-7) was associated with lower AD risk. Four out of seven traits based on a survey about education and qualifications were significantly associated with AD (*Figure 3—figure supplement 2*). Higher education such as having a university degree (p=4.4e-12), A levels/AS levels or equivalent (p=6.9e-9), and professional qualifications (p=7.1e-6) were associated with lower AD risk. In contrast, choosing 'none of the above' in this survey was associated with a higher risk (p=1.6e-11). Other notable strong associations include high cholesterol (p=2.5e-15; positive), lifestyle traits such as cheese intake (p=2.5e-10; negative), occupation traits such as job involving heavy physical work (p=2.7e-10; positive), anthropometric traits including height (p=5.3e-7; negative), and traits related to pulmonary function, e.g., forced expiratory volume in 1 s (FEV1; p=1.9e-6; negative). Detailed information on all associations is summarized in *Supplementary file 2*.

## Multivariate conditional analysis identifies independently associated risk factors

Of note, associations identified in the marginal analysis are not guaranteed to be independent. We observed clear correlational structures among the identified traits (*Figure 4*). For example, PRS of various intelligence and cognition-related traits are strongly correlated, and consumption of cholesterol-lowering medication is correlated with self-reported high cholesterol. To account for the correlations among traits and identify risk factors that are independently associated with AD, we performed multivariate conditional analysis using GWAS summary statistics (**Methods**). First, we applied hierarchical clustering to the 48 traits we identified in marginal association analysis and divided these traits into 15 representative clusters. The traits showing the most significant marginal association in each cluster were included in the multivariate analysis (*Figure 4—figure supplement 1*). Similar to the marginal analysis, we analyzed IGAP and ADGC data separately and combined the results using meta-analysis (*Supplementary file 2*). All 15 representative traits remained nominally significant (p<0.05) and showed consistent effect directions between marginal and conditional analyses (*Supplementary file 1*). However, several traits showed substantially reduced effect estimates and inflated p-values in multivariate analysis, including fluid intelligence score, mother still alive, unable to work because of sickness or disability, duration of moderate activity, and intake of cholesterol-lowering spread. Interestingly, major trait categories that showed the strongest marginal associations with AD (i.e. family history, high cholesterol, and education/cognition) were independent from each other. Paternal and maternal family history also showed independent associations with AD, consistent with the low correlation between their genetic risk scores (correlation = 0.052).

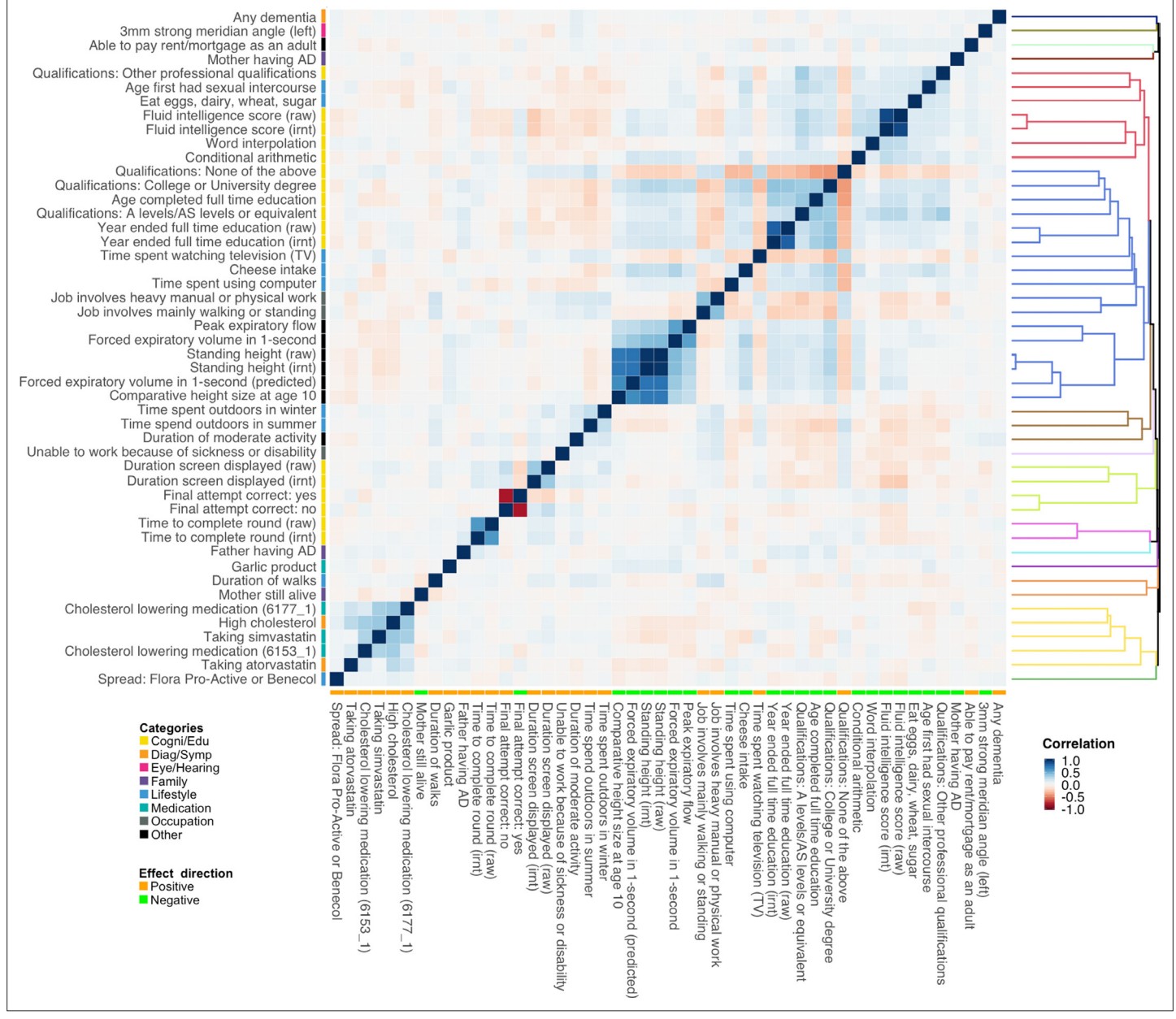

**Figure 4.** Polygenic risk score (PRS) correlation matrix for the 48 traits identified in marginal association analysis. Trait categories and association directions with Alzheimer's disease (AD) are annotated. The dendrogram indicates the results of hierarchical clustering. We used 1000 genome samples with European ancestry to calculate PRS and evaluate their correlations. Label 'irnt' means that trait values were standardized using rank-based inverse normal transformation in the genome-wide association study (GWAS) analysis.

The online version of this article includes the following figure supplement(s) for figure 4:

**Figure supplement 1.** Correlation heatmap for the 15 representative traits selected based on hierarchical clustering.

## Influence of the *APOE* region on identified associations

Furthermore, we evaluated the impact of *APOE* on identified associations. We removed the extended *APOE* region (chr19: 45,147,340–45,594,595; hg19) from summary statistics of the 48 traits showing significant marginal associations with AD and repeated the analysis. We observed a substantial drop in the significance level of many traits, especially family history of AD, dementia diagnosis, and high cholesterol (*Figure 5*, *Figure 5—figure supplement 1*, and *Supplementary file 2*). 38 out of 48 traits remained significant under stringent Bonferroni correction after *APOE* removal. Interestingly,

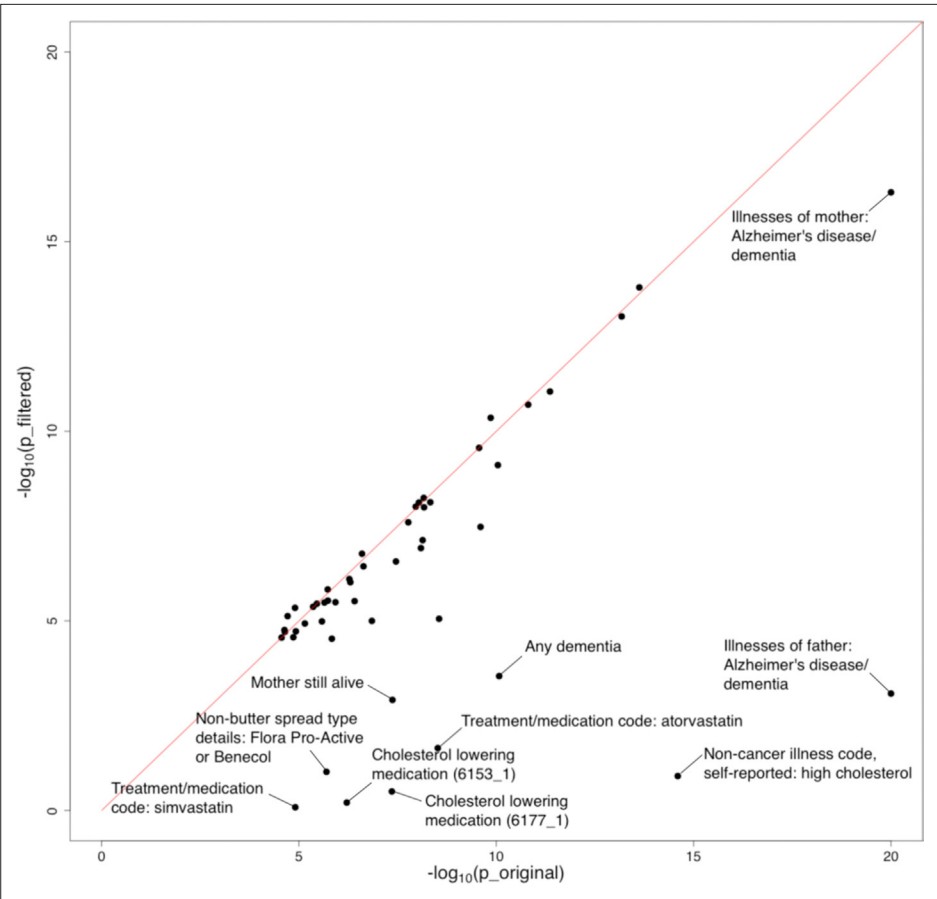

**Figure 5.** Influence of the *APOE* region on trait-Alzheimer's disease (AD) associations. The horizontal and vertical axes denote association p-values before and after removal of the *APOE* region, respectively. Original p-values (i.e. the x-axis) were truncated at 1e-20 for visualization purposes.

The online version of this article includes the following figure supplement(s) for figure 5:

**Figure supplement 1.** Influence of a wider *APOE* region on polygenic risk score (PRS)-Alzheimer's disease (AD) associations.

the associations between AD and almost all cognition/intelligence traits were virtually unchanged, suggesting a limited role of *APOE* in these associations.

## Causal inference via Mendelian randomization

Next, we investigated the evidence for causality among identified associations. We performed Mendelian randomization (MR-IVW; **Methods**) in IGAP and ADGC datasets separately and meta-analyzed the results on the complete set of 1738 heritable traits from the UK biobank. A total of 48 traits reached Bonferroni-corrected statistical significance and showed consistent effect directions in the discovery and replication analyses using BADGERS. In contrast, MR-IVW only identified nine traits with Bonferroni-corrected statistical significance. Among these nine traits, seven were also identified by BADGERS (*Supplementary file 1*). The signs of all significant causal effects identified by MR-IVW were consistent with results from BADGERS. The most significant effect was family history (p=1.1e-233 and 1.7e-69 for maternal and paternal history, respectively). Dementia diagnosis (p=9.1e-7), high cholesterol (p=4.1e-6), A levels/AS levels education (p=1.7e-4), and time spent watching television (p=2.4e-4) were also among the top significant effects. Of note, the fluid intelligence score, one of the most significant associations identified by BADGERS, did not reach statistical significance in MR (p=0.06), which may be explained by its polygenic genetic architecture. It is also worth noting that if we scan all 1738 traits using BADGERS and then apply MR-IVW on the 48 Bonferroni-corrected

significant traits, 23 could reach nominal significance (p<0.05) in MR, and seven could reach significance under Bonferroni correction (p<0.05/48; *Supplementary file 1*).

We also compared BADGERS with another more recent method GSMR (*Zhu et al., 2018*). Due to the smaller sample size in the ADGC dataset, we only applied GSMR to the IGAP summary statistics. In total, 18 traits reached statistical significance under Bonferroni correction (*Supplementary file 1*). However, these results showed only moderate consistency with MR-IVW and BADGERS. Among the 18 significant traits, only 1 trait, maternal family history of Alzheimer's disease and dementia, overlapped with significant traits identified by MR-IVW. Six out of 18 traits overlapped with significant traits identified by BADGERS. Among the 18 significant traits, eight are related to body fat mass and two are related to educational attainment. The most significant effect was illnesses of mother (p=2.4e-294). College or University degree (p=4.84e-6), education; none of the above (p=3.6e-4), A levels/AS levels education (p=3.8e-6), and time spent watching television (p=4.0e-3) were also among top significant effects. Notably, GSMR failed to identify paternal family history or high cholesterol as risk factors for Alzheimer's disease. If we only consider the 48 significant traits identified by BADGERS, 11 were nominally significant (p<0.05). However, 23 traits did not have enough significant SNPs to perform the GSMR analysis (at least 10 SNPs are required). The signs of all significant causal effects identified by GSMR were consistent between association effects in BADGERS.

Additionally, we included GSMR analysis results after removing *APOE* region from the 48 identified traits. Only maternal family history reached Bonferroni-corrected statistical significance, further demonstrating the lack of statistical power in MR when performing biobank-wide scans (*Supplementary file 1*).

## Associations with AD subgroups, biomarkers, and pathologies

To further investigate the mechanistic pathways for the identified risk factors, we applied BADGERS to a variety of AD subgroups, biomarkers, and neuropathologic features (*Supplementary file 1*). Overall, 29 significant associations were identified under a false discovery rate (FDR) cutoff of 0.05, and these endophenotypes showed distinct association patterns with AD risk factors (*Figure 6*; *Figure 6—figure supplement 1*). First, we tested the associations between the 48 AD-associated traits and five AD subgroups defined in the Executive Prominent Alzheimer's Disease (EPAD) study, i.e., memory, language, visuospatial, none, and mix (**Methods**) (*Mukherjee et al., 2018*; *Crane et al., 2017*). Maternal family history of AD and dementia was strongly and consistently associated with all five EPAD subgroups (*Supplementary file 2*), with memory subgroup showing the strongest association (p=3.3e-16), which is consistent with the higher frequency of *APOE* ε4 in this subgroup (*Mukherjee et al., 2018*). Paternal family history was not strongly associated with any subgroups, but the effect directions were consistent. Interestingly, intelligence and cognition-related traits such as correct final attempt in cognitive test (p=2.7e-5) and fluid intelligence score (p=6.8e-5) were specifically associated with the 'none' subgroup – AD samples without relative impairment in any of the four cognitive domains. High cholesterol and related traits were associated with language, memory, and mix (i.e. AD samples with relative impairment in two or more domains) subgroups but showed weaker associations with the visuospatial and none subgroups.

Next, we extended our analysis to three biomarkers of AD in cerebrospinal fluid (CSF): amyloid beta ($A\beta_{42}$), tau, and phosphorylated tau ($ptau_{181}$) (*Deming et al., 2017*). Somewhat surprisingly, AD risk factors did not show strong associations with $A\beta_{42}$ and tau (*Supplementary file 2*). Maternal family history of AD and dementia was associated with $ptau_{181}$ (p=4.2e-4), but associations were absent for $A\beta_{42}$ and tau. It has been recently suggested that CSF biomarkers have a sex-specific genetic architecture (*Deming et al., 2018*). However, no association passed an FDR cutoff of 0.05 in our sex-stratified analyses (*Supplementary file 2*).

Furthermore, we applied BADGERS to a variety of neuropathologic features of AD and related dementias (**Methods**), including neuritic plaques (NPs), neurofibrillary tangles (NFTs), cerebral amyloid angiopathy (CAA), lewy body disease (LBD), hippocampal sclerosis (HS), and vascular brain injury (VBI) (*Beecham et al., 2014*). Family history of AD/dementia (p=3.8e-8, maternal; p=1.4e-5, paternal) and high cholesterol (p=2.1e-5) were strongly associated with NFT Braak stages (*Supplementary file 2*). NP also showed very similar association patterns with these traits (p=2.7e-19, maternal family history; p=2.6e-7, paternal family history; p=0.001, high cholesterol). The other neuropathologic features did not show strong associations. Of note, despite not being statistically significant, family history of AD/

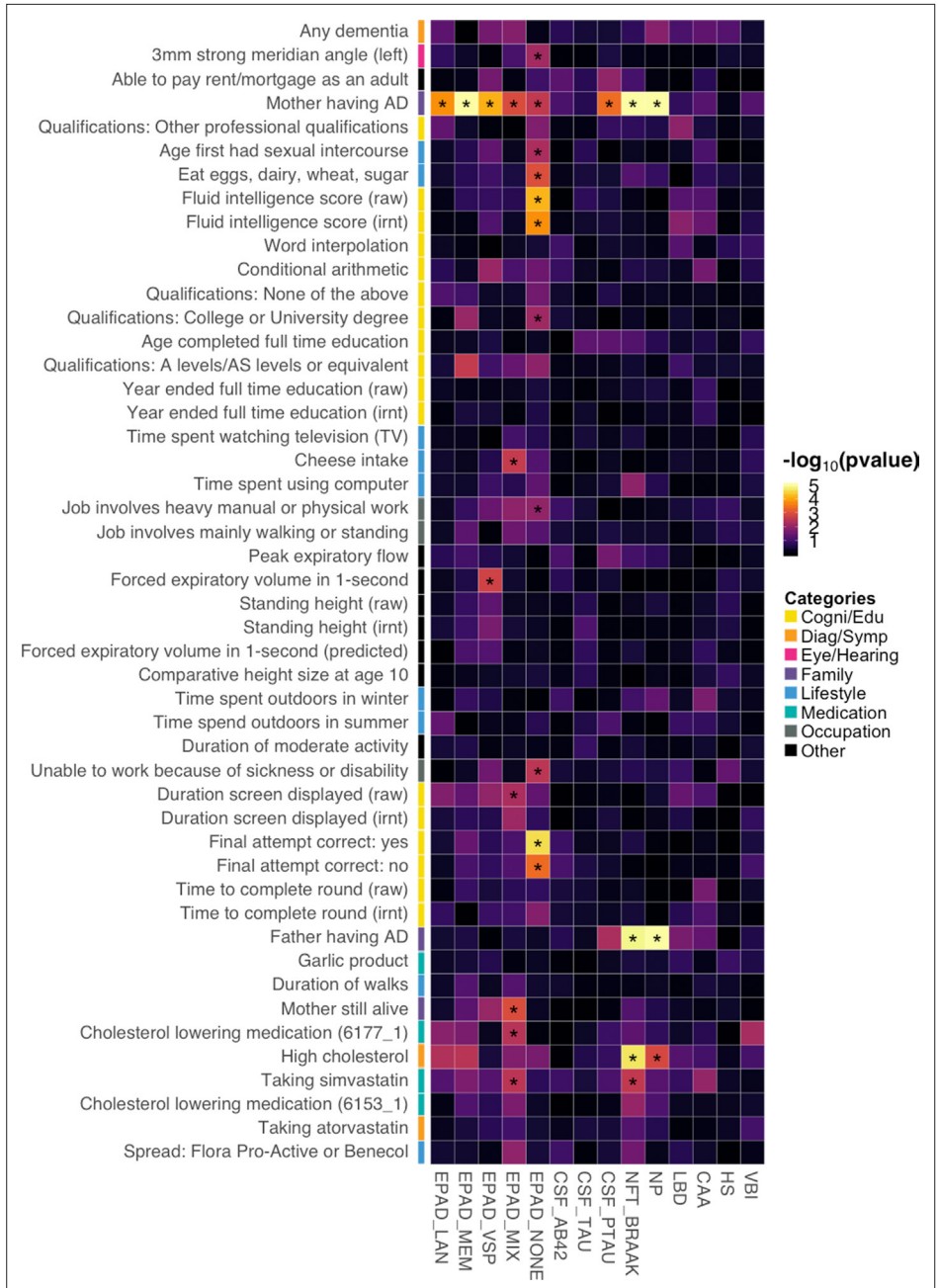

**Figure 6.** Associations between identified Alzheimer's disease (AD) risk factors and various AD subgroups, cerebrospinal fluid (CSF) biomarkers, and neuropathologic features. Asterisks denote significant associations based on an false discovery rate (FDR) cutoff of 0.05. p-values are truncated at 1e-5 for visualization purposes.

The online version of this article includes the following figure supplement(s) for figure 6:

**Figure supplement 1.** Association directions between identified Alzheimer's disease (AD) risk factors and AD endophenotypes.

**Figure supplement 2.** Association results for the complete set of 13 neuropathologic features for Alzheimer's disease (AD) and other dementias.

dementia was negatively associated with VBI, and multiple intelligence traits were positively associated with LBD, showing distinct patterns with other pathologies (*Figure 6—figure supplement 2*). We also note that various versions of the same pathologies all showed consistent associations in our analyses (*Figure 6—figure supplement 2*). The complete association results for all the endophenotypes

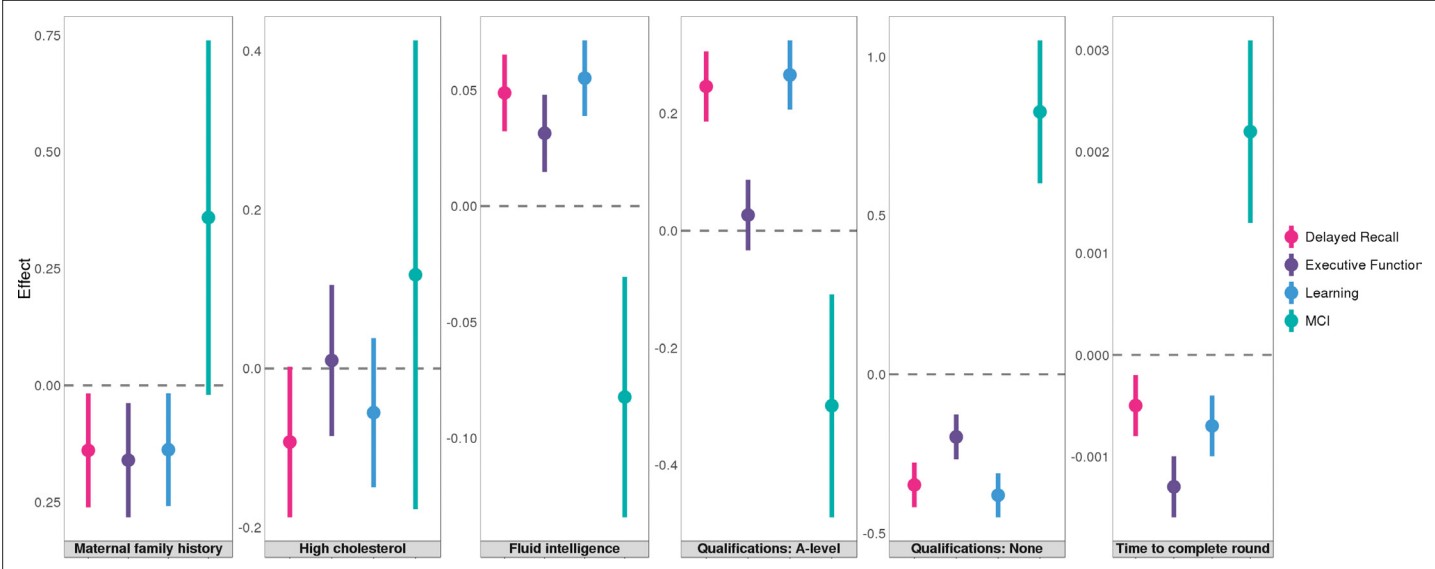

**Figure 7.** Associations between six traits and pre-clinical cognitive phenotypes in Wisconsin Registry for Alzheimer's Prevention (WRAP). Error bars denote the standard error of effect estimates. N=1,198.

and all the traits are summarized in *Supplementary file 2*. We further identified the influence of the *APOE* region in these results. The association results for all the endophenotypes with *APOE* Region being removed are summarized in *Supplementary file 2*.

## Associations with cognitive traits in a pre-clinical cohort

Finally, we studied the associations between AD risk factors and pre-clinical cognitive phenotypes using 1198 samples from the Wisconsin Registry for Alzheimer's Prevention (WRAP), a longitudinal study of initially dementia-free middle-aged adults (*Johnson et al., 2018*). Assessed phenotypes include mild cognitive impairment (MCI) status and three cognitive composite scores for executive function, delayed recall, and learning (**Methods**). A total of 12 significant associations reached an FDR cutoff of 0.05 (*Supplementary file 2*). Somewhat surprisingly, parental history and high cholesterol, the risk factors that showed the strongest associations with various AD endophenotypes, were not associated with MCI or cognitive composite scores in WRAP. Instead, education and intelligence-related traits strongly predicted pre-clinical cognition (*Figure 7*). A-levels education and no education both showed highly significant associations with delayed recall (p=4.0e-5 and 7.7e-7) and learning (p=7.6e-6 and 5.0e-8). No education was also associated with higher risk of MCI (p=2.5e-4). Additionally, fluid intelligence score was positively associated with the learning composite score (p=7.5e-4), and time to complete round in cognition test was negatively associated with the executive function (p=1.1e-5).

## Discussion

In this work, we introduced BADGERS, a new method to perform association scans at the biobank scale using genetic risk scores and GWAS association statistics. Through simulations, we demonstrated that our method provides consistent effect estimates and similar statistical power compared to regression analysis based on individual-level data. Additionally, we applied BADGERS to two large and independent GWAS datasets for late-onset AD. In our analyses, we used GWAS summary statistics from the UK biobank, one of the largest genetic cohort in the world, to generate PRS for complex traits. We estimated heritability for all available traits in the UK biobank and only included traits with nominally significant heritability (p<0.05) in our analyses. The GWAS summary statistics for Alzheimer's disease were also obtained from the largest available study – International Genomics of Alzheimer's Project (IGAP) and we further sought replication using a large, independent dataset from the Alzheimer's Disease Genetics Consortium (ADGC). Overall, we are confident that these quality

control procedures largely controlled the false findings in our study. Among 1738 heritable traits in the UK biobank, we identified 48 traits showing statistically significant associations with AD. These traits covered a variety of categories, including family history, cholesterol, intelligence, education, occupation, and lifestyle. Although many of the identified traits are genetically correlated, multivariate conditional analysis confirmed multiple strong and independent associations for AD. Family history showing strong associations with AD is not a surprise, and many other associations are supported by the literature as well. The protective effect of higher educational and occupational attainment on the risk and onset of dementia is well studied (*Valenzuela and Sachdev, 2006*; *Stern, 2012*). Cholesterol buildup is also known to associate with β-amyloid plaques in the brain and higher AD risk (*Reed et al., 2014*; *Djelti et al., 2015*; *Simons et al., 2001*).

More interestingly, these identified traits had distinct association patterns with various AD subgroups, biomarkers, pathologies, and pre-clinical cognitive traits. Five cognitively-defined AD subgroups were consistently associated with maternal family history, but only the group without substantial relative impairment in any domain (i.e. EPAD_none) was associated with intelligence and education. In addition, family history and high cholesterol were strongly associated with classic AD neuropathologies, including NP and NFT, while intelligence and educational attainment predicted pre-clinical cognitive scores and MCI. These results suggest that various AD risk factors may affect the disease course at different time points and via distinct biological processes, and genetically predicted risk factors for clinical AD include at least two separate components. While some risk factors (e.g. high cholesterol and *APOE*) may directly contribute to the accumulation of pathologies, other factors (e.g. intelligence and education) may buffer the adverse effect of brain pathology on cognition (*Stern, 2012*). One possible scenario is that family history and high cholesterol are the fundamental causes of AD while education level and intelligence are the parameters of such factors. While if one didn't have such a factor in the first stage, they are protected from getting AD, if someone with such factor and also has high score in education attainment or intelligence, they can also get rid of the possibility of getting AD. We also investigated the influence of *APOE* on the identified associations. Effects of family history and high cholesterol were substantially reduced after *APOE* removal. In contrast, associations with cognition and education were virtually unchanged. These results suggest that various AD risk factors may affect the disease course at different time points and via distinct biological processes. While some risk factors (e.g. high cholesterol and *APOE*) may directly contribute to the accumulation of pathologies, other factors (e.g. intelligence and education) reduce the adverse effect of brain pathology on cognition (*Stern, 2012*).

Furthermore, we note that the association results in BADGERS need to be interpreted with caution. Although PRS-based association analysis is sometimes treated as causal inference in the literature (*Paternoster et al., 2017*), we do not see BADGERS as a tool to identify causal factors. Key assumptions in causal inference are in many cases, violated when analyzing complex, highly polygenic traits, which may lead to complications when interpreting results. In our analysis, BADGERS showed superior statistical power than MR-IVW – among 1738 heritable traits, 48 reached Bonferroni significance in BADGERS, 9 and 18 traits reached Bonferroni significance in MR-IVW and GSMR, respectively. Among the 48 traits identified by BADGERS, 23 reached nominal statistical significance in MR-IVW and 11 were nominally significant in GSMR. BADGERS is a statistically powerful and computationally efficient method for identifying associations between a disease of interest and genetically imputed complex traits. Due to the capability of utilizing PRS with a large number of SNPs to impute complex traits, BADGERS has substantially improved statistical power compared to MR methods. And because of this, it can serve as a hypothesis-free method to screen for candidate risk factors from biobank-scale datasets with an overwhelming number of traits. After a list of candidate risk factors is identified using BADGERS, MR methods can be applied to carefully demonstrate causality. We envision BADGERS as a tool to prioritize associations among a large number of candidate risk factors so that robust causal inference methods can be applied to carefully assess causal effects. In addition, BADGERS requires a reference panel to provide LD estimates as a summary statistics-based method. If the population in the reference panel does not match that of the GWAS, it may create bias in the analysis. Our simulation results suggest that 1000 Genomes European samples is sufficient for our analysis when the GWAS was also conducted on European samples. Our implemented BADGERS software is flexible on the choice of LD reference panel. It allows users to change the reference dataset when they see fit.

What's more, environmental factors may play a big role in the identified associations. There is little doubt that the environment could influence many complex traits, including the ones highlighted by the reviewer. However, this does not necessarily mean that these traits cannot also have a genetic component (or be genetically heritable). we summarized the heritability estimates for the 48 traits identified in our BADGERS meta-analysis of two independent datasets for Alzheimer's disease (*Supplementary file 1*), and all of them have nominally significant heritability estimates (p<0.05) based on our selection criteria. Nevertheless, we do acknowledge that the high heritability of these traits is influenced by correlations with other traits. For example, job involving heavy manual or physical work is genetically correlated with educational attainment (*Figure 3*), which indicates that the association between this trait and Alzheimer's disease may not be direct. Therefore, it is important to note that association results from BADGERS analysis need to be interpreted with caution.

Limited sample size in AD endophenotypes is another limitation in our study. We have used data from the largest available GWAS for CSF biomarkers and neuropathologies. Still, the small sample size made it challenging to assess the effects of traits that were weakly associated with AD. When an independent validation dataset is available, it would be of interest to assess the prediction accuracy of PRS on each trait. However, external validation datasets rarely exist in real applications. In that case, the users may choose to use heritability estimates to filter traits with a substantial genetic component. Furthermore, in the BADGERS framework, PRS are independent variables in the regression analysis. If the PRS has limited predictive power, such noise is similar to measurement errors in standard regression analysis. This may decrease the statistical power in association tests but does not inflate the type-I error rate. Finally, emerging evidence has highlighted the sex-specific genetic architecture of AD (*Deming et al., 2018*; *Hohman et al., 2018*). In our analysis, maternal family history of AD showed stronger associations with various phenotypes than paternal family history. However, we note that this may be explained by the sample size difference in the UK biobank ($N_{case}$ = 28,507 and 15,022 for samples with maternal and paternal family history, respectively). We also performed sex-stratified analyses for CSF biomarkers but identified limited associations, possibly due to the small sample size. Overall, sex-specific effects of risk factors remain to be investigated in the future using larger datasets. In total, BADGERS requires the training data for genetic prediction models and the downstream disease GWAS to be independent but of similar genetic ancestry. Development of methods that are more robust to sample overlap and diverse genetic ancestry remains an open problem for future research.

In conclusion, BADGERS is a statistically powerful method to identify associated risk factors for complex diseases. Large-scale biobanks continue to provide rich data on various human traits that may be of interest in disease research. Our method uses GWAS to bridge large biobanks with studies on specific diseases, lessens the limitation of insufficient disease cases in biobanks and lack of risk factor measurements in disease studies, and provides a statistically justified approach to identifying risk factors for disease. We have demonstrated the effectiveness of BADGERS through extensive simulations, a two-stage BWAS for late-onset AD, and various follow-up analyses on identified risk factors. Our results provided new insights into the genetic basis of AD, and revealed distinct mechanisms for the involvement of risk factors in AD etiologies. The ever-growing sample size in GWAS and biobanks, in conjunction with increasingly accessible summary association statistics, makes BADGERS a powerful and valuable tool in human genetics research.

## Methods
### BADGERS framework

The goal of this method is to study the association between $Y$, a measured trait in the study, and $\hat{T}$, a trait imputed from genetic data via a linear prediction model:

$$\hat{T} = XW$$

Here, $X_{N \times M}$ is the genotype matrix for $N$ individuals in a study of trait $Y$. $W_{M \times 1}$ is the pre-calculated weight values on SNPs in the imputation model. $M$ denotes the number of SNPs. We use $Y$, a $N \times 1$ vector, to denote the trait values measured on the same group of individuals. We test the association between $Y$ and $\hat{T}$ via a linear model.

$$Y = \alpha + \hat{T}\gamma + \delta$$

where $\alpha$ is the intercept, $\delta$ is the term for random noise, and regression coefficient $\gamma$ is the parameter of interest. The ordinary least squares (OLS) estimator for $\gamma$ can be denoted as,

$$\hat{\gamma} = \frac{cov\left(\hat{T}, Y\right)}{var\left(\hat{T}\right)} = \frac{cov\left(XW, Y\right)}{var\left(\hat{T}\right)} = \frac{1}{var\left(\hat{T}\right)} W^T \begin{pmatrix} cov\left(X_1, Y\right) \\ \vdots \\ cov\left(X_M, Y\right) \end{pmatrix}$$

Here, $X_j$ is the j$^{th}$ column of $X$. Additionally, we derive the formula for the standard error of $\hat{\gamma}$ :

$$se\left(\hat{\gamma}\right) = \sqrt{\frac{var\left(\delta\right)}{N \times var\left(\hat{T}\right)}} \approx \sqrt{\frac{var\left(Y\right)}{N \times var\left(\hat{T}\right)}}$$

The approximation in this formula is based on the assumption that trait $Y$ has complex etiology and imputed trait $\hat{T}$ only explains a small proportion of its phenotypic variance. When an accurate estimate of $var\left(\delta\right)$ is difficult to obtain, this approximation approach provides conservative results and controls type-I error in the analysis.

In practice, individual-level genotype (i.e. $X$) and phenotype data (i.e. $Y$) may not be accessible due to policy and privacy concerns. Therefore, it is of practical interest to perform the aforementioned association analysis using summary association statistics. Standard genetic association analysis tests the association between trait $Y$ and each SNP via the following linear model:

$$Y = \mu_j + X_j\beta_j + \varepsilon_j \left(1 \leq j \leq M\right)$$

The OLS estimator for $\beta_j$ and its standard error have the following forms.

$$\hat{\beta}_j = \frac{cov\left(X_j, Y\right)}{var\left(X_j\right)}$$

$$se\left(\hat{\beta}_j\right) = \sqrt{\frac{var\left(\varepsilon_j\right)}{N \times var\left(X_j\right)}} \approx \sqrt{\frac{var\left(Y\right)}{N \times var\left(X_j\right)}}$$

Again, the approximation is based on the empirical observation in complex trait genetics – each SNP explains little variability of $Y$ (**Manolio et al., 2009**).

Next, we derive the test statistic (i.e. z-score) for $\gamma$:

$$Z = \frac{\hat{\gamma}}{se(\hat{\gamma})}$$

$$\approx \sqrt{\frac{N}{var(Y) \times var(\hat{T})}} W^T \begin{pmatrix} cov(X_1, Y) \\ \vdots \\ cov(X_M, Y) \end{pmatrix}$$

$$\approx \sqrt{\frac{1}{var(\hat{T})}} W^T \begin{pmatrix} \frac{\sqrt{var(X_1)}\hat{\beta}_1}{se(\hat{\beta}_1)} \\ \vdots \\ \frac{\sqrt{var(X_M)}\hat{\beta}_M}{se(\hat{\beta}_M)} \end{pmatrix}$$

$$= W^T \Gamma \widetilde{Z}$$

where $\Gamma$ is a diagonal matrix with the j$^{th}$ diagonal element being

$$\Gamma_{jj} = \sqrt{\frac{var\left(X_j\right)}{var\left(\hat{T}\right)}}$$

and $\widetilde{Z}$ is the vector of SNP-level z-scores obtained from the GWAS of trait $Y$, i.e.,

$$\widetilde{Z}_j = \frac{\hat{\beta}_j}{se\left(\hat{\beta}_j\right)}$$

Without access to individual-level genotype data, $var\left(X_j\right)$ and $var(\hat{T})$ need to be estimated using an external panel with a similar ancestry background. We use $\widetilde{X}$ to denote the genotype matrix from an external cohort, then $var\left(X_j\right)$ can be approximated using the sample variance of $\widetilde{X}_j$. Variance of $\hat{T}$ can be approximated as follows

$$var\left(\hat{T}\right) \approx W^T \widetilde{D} W$$

where $\widetilde{D}$ is the variance-covariance matrix of all SNPs estimated using $\widetilde{X}$. However, when the number of SNPs is large in the imputation model for trait $T$, calculation of $\widetilde{D}$ is computationally intractable. Instead, we use an equivalent but computationally more efficient approach. We first impute trait $T$ in the external panel using the same imputation model

$$\widetilde{T} = \widetilde{X} W$$

Then, $var(\hat{T})$ can be approximated by sample variance $var(\widetilde{T})$.

Thus, we can test the association between $Y$ and $\hat{T}$ without having access to individual-level genotype and phenotype data from the GWAS. The required input variables for BADGERS include a linear imputation model for trait $T$, SNP-level summary statistics from a GWAS of trait $Y$, and an external panel of genotype data. With these, the association test can be performed.

## Multivariate analysis in BADGERS

To adjust for potential confounding effects, it may be of interest to include multiple imputed traits in the same BADGERS model. We still use $Y$ to denote the measured trait of interest. The goal is to perform a multiple regression analysis using $K$ imputed traits (i.e. $\hat{T}_1$,..., $\hat{T}_K$) as predictor variables:

$$Y = \hat{T}^* \gamma^* + \delta^*$$

Here, we use $\hat{T}^* = (\hat{T}_1, \ldots, \hat{T}_K)$ to denote a $N \times K$ matrix for $K$ imputed traits. Regression coefficients $\gamma^* = \left(\gamma_1, \ldots, \gamma_K\right)^T$ are the parameters of interest. To simplify algebra, we also assume trait $Y$ and all SNPs in the genotype matrix $X$ are centered so there is no intercept term in the model, but the conclusions apply to the general setting. Similar to univariate analysis, traits $\hat{T}_1, \ldots, \hat{T}_K$ are imputed from genetic data via linear prediction models:

$$\hat{T}^* = X W^*$$

where $W^*_{M \times K}$ are imputation weights assigned to SNPs. The $i^{th}$ column of $W$ denotes the imputation model for trait $T_i$. Then, the OLS estimator $\hat{\gamma}^*$ and its variance-covariance matrix can be denoted as follows:

$$\hat{\gamma}^* = \left(\left(\hat{T}^*\right)^T \hat{T}^*\right)^{-1} \left(\hat{T}^*\right)^T Y$$

$$cov\left(\hat{\gamma}^*\right) \approx var\left(Y\right) \left(\left(\hat{T}^*\right)^T \hat{T}^*\right)^{-1}$$

The approximation is based on the assumption that imputed traits $\hat{T}_1, \ldots, \hat{T}_K$ collectively explain little variance in $Y$, which is reasonable in complex trait genetics if $K$ is not too large. We further denote:

$$\mathbf{U} := N\left(\left(\hat{T}^*\right)^T \hat{T}^*\right)^{-1} = \begin{pmatrix} var\left(\hat{T}_1\right) & \cdots & cov\left(\hat{T}_1, \hat{T}_K\right) \\ \vdots & \ddots & \vdots \\ cov\left(\hat{T}_K, \hat{T}_1\right) & \cdots & var\left(\hat{T}_K\right) \end{pmatrix}^{-1}$$

All elements in matrix $U$ can be approximated using a reference panel $\widetilde{X}$ (**Dudbridge, 2013**):

$$cov\left(\hat{T}_i, \hat{T}_j\right) \approx cov\left(\widetilde{T}_i, \widetilde{T}_j\right)$$

Therefore, the z-score for $\gamma_k$ $(1 \le k \le K)$ is

$$\begin{aligned} Z_k &= \frac{\hat{\gamma_k}}{se(\hat{\gamma_k})} \\ &= \frac{I_k^T U(W^*)^T X^T Y}{\sqrt{N U_{kk} var(Y)}} \\ &= \frac{1}{\sqrt{U_{kk}}} I_k^T U\left(W^*\right)^T \Theta \widetilde{Z} \end{aligned}$$

where $I_k$ is the $K \times 1$ vector with the k$^{th}$ element being 1 and all other elements equal to 0, is a $M \times M$ diagonal matrix with the i$^{th}$ diagonal element being $\sqrt{var\left(X_i\right)}$, and similar to the notation in univariate analysis, $\widetilde{Z}$ is the vector of SNP-level z-scores from the GWAS of trait $Y$. Given imputation models for $K$ traits (i.e. $W^*$), GWAS summary statistics for trait $Y$ (i.e. $\widetilde{Z}$), and an external genetic dataset to estimate $U$ and , multivariate association analysis can be performed without genotype and phenotype data from the GWAS.

## Genetic prediction

Any linear prediction model can be used in the BADGERS framework. With access to individual-level genotype and phenotype data, the users can train their preferred statistical learning models, e.g., penalized regression or linear mixed model. When only GWAS summary statistics are available for risk factors (i.e. $T$), PRS can be used for imputation. We used PRS to impute complex traits in all analyses throughout the paper. Of note, more advanced PRS methods that explicitly model LD (**Vilhjálmsson et al., 2015**) and functional annotations (**Hu et al., 2017**) to improve prediction accuracy have been developed. However, additional independent datasets may be needed if there are tuning parameters in PRS. In general, higher imputation accuracy will improve statistical power in association testing (**Hu et al., 2018**). The BADGERS software allows users to choose their preferred imputation model.

## Simulation settings

We simulated quantitative traits using genotype data of 62,313 individuals from the GERA cohort (dbGap accession: phs000674). Summary association statistics were generated using PLINK (**Purcell et al., 2007**). We ran BADGERS on summary statistics based on the simulated traits and PRS of 1738 traits in the UK biobank. To compare BADGERS with the traditional approach that uses individual-level data as input, we also directly regressed simulated traits on the PRS of UK biobank traits to estimate association effects.

### Setting 1

We simulated quantitative trait values as i.i.d. samples from normal distribution with mean 0 and variance 1. In this setting, simulated trait values were independent from genotype data.

### Setting 2

We simulated quantitative trait values based on an additive random effect model commonly used in heritability estimation (**Yang et al., 2015**). We fixed heritability to be 0.1. In this setting, the simulated trait is associated with SNPs, but is not directly related to PRS of UK biobank traits.

## Setting 3

We selected 100 traits from 1738 UK-Biobank traits to calculate PRS on GERA data. For each of these 100 PRS, we simulated a quantitative trait by summing up the effect of PRS, a polygenic genetic background, and a noise term.

$$Y = X\beta + \rho P + \varepsilon$$

Here, $X$ denotes the genotype of samples; $\beta$ is the effect size of each variant; $P$ is the PRS of one of the selected traits; $\rho$ is the effect size of PRS; and $\varepsilon$ is the error term following a standard normal distribution. The polygenic background and random noise (i.e. $X\beta + \varepsilon$) were simulated using the same model described in setting 2. This term and the PRS were normalized separately. The standardized effect size (i.e. $\rho$) was set as 0.02, 0.015, 0.01, 0.008, and 0.005 in our simulations. In this setting, simulated traits are directly associated with SNPs and PRS. For each value of $\rho$, statistical power was calculated as the proportion of significant results (p<0.05) out of 100 traits.

## Setting 4

We simulated 100 quantitative traits $T_1, \ldots, T_{100}$ based on an additive random effect model commonly used with heritability fixed as 0.1. And the response traits $Y_1, \ldots, Y_{100}$ were simulated by adding a noise term to $T$.

$$Y_i = \gamma_i T_i + \varepsilon_i$$

Where $\gamma_i \sim N(0, 2)$, and $\varepsilon_i \sim N(0, Var(T_i))$. The dataset was split into two subsets, one with 31,162 (subset 1) and another with 31,163 samples (subset 2). Marginal summary statistics correspond to $T_i$'s and $Y_i$'s were derived using subset 1 and subset 2, respectively. We applied LDpred to jointly estimate all SNPs' effects using marginal summary statistics from subset 1. Then, we ran BADGERS to identify associations between 100 pairs of $Y_i$ and $T_i$ using two methods to impute $T_i$'s (i.e. marginal PRS and LDpred).

## GWAS datasets

Summary statistics for 4357 UK biobank traits were generated by Dr. Benjamin Neale's group and were downloaded from (http://www.nealelab.is/uk-biobank). AD summary statistics from the IGAP stage-I analysis were downloaded from the IGAP website (http://web.pasteur-lille.fr/en/recherche/u744/igap/igap_download.php). ADGC phase 2 summary statistics were generated by first analyzing individual datasets using logistic regression adjusting for age, sex, and the first three principal components in the program SNPTest v2 (*Marchini et al., 2007*). Meta-analysis of the individual dataset results was then performed using the inverse-variance weighted approach (*Willer et al., 2010*).

GWAS summary statistics for neuropathologic features of AD and related dementias were obtained from the ADGC. Details on these data have been previously reported (*Beecham et al., 2014*). We analyzed a total of 13 neuropathologic features, including four NP traits, two traits for NFT Braak stages, three traits for LBD, CAA, HS, and two VBI traits. Among different versions of the same pathology, we picked one dataset for each pathologic feature to show in our primary analyses. Six AD subgroups were defined in the recent EPAD paper (*Mukherjee et al., 2018*) on the basis of relative performance in memory, executive functioning, visuospatial functioning, and language at the time of Alzheimer's diagnosis. Four subgroups include AD samples with an isolated substantial relative impairment in one of four domains; the 'none' subgroup includes samples without substantial relative impairment; the 'mix' subgroup includes samples with relative impairment in multiple domains. Each domain was compared with healthy controls in case-control association analyses. We did not include the executive functioning subgroup in our analysis due to its small sample size in cases. Detailed information about the design of CSF biomarker GWAS and the recent sex-stratified analysis has been described previously (*Deming et al., 2017*; *Deming et al., 2018*). Details on the association statistics for AD subgroups, CSF biomarkers, and neuropathological features are summarized in *Supplementary file 2*.

## Analysis of GWAS summary statistics

We applied LD score regression implemented in the LDSC software (*Bulik-Sullivan et al., 2015*) to estimate the heritability of each trait. Among 4357 traits, we selected 1738 with nominally significant heritability (p<0.05) to include in our analyses. We removed SNPs with association p-values greater than 0.01 from each of the 1738 summary statistics files, clumped the remaining SNPs using a LD cutoff of 0.1 and a radius of 1 Mb in PLINK (*Purcell et al., 2007*), and built PRS for each trait using the effect size estimates of remaining SNPs.

Throughout the paper, we used samples of European ancestry in the 1000 Genomes Project as a reference panel to estimate LD (*Abecasis et al., 2012*). In univariate analyses, we tested marginal associations between each PRS and AD using the IGAP stage-I dataset and replicated the findings using the ADGC summary statistics. Association results in two stages were combined using an inverse variance-weighted meta-analysis (*Willer et al., 2010*). A stringent Bonferroni-corrected significance threshold was used to identify AD-associated risk factors. For associations between identified risk factors and AD endophenotypes, we used an FDR cutoff of 0.05 to claim statistical significance. We applied hierarchical clustering to the covariance of 48 traits we identified from marginal association analysis, then divided the result into 15 clusters and selected one most significant trait from each cluster and used them to perform multivariate conditional analysis. We analyzed IGAP and ADGC datasets separately, and combined the results using meta-analysis.

We used MR-IVW approach (*Burgess et al., 2013*) implemented in the Mendelian Randomization R package (*Yavorska and Burgess, 2017*) to study the causal effects of 48 risk factors identified by BADGERS. For each trait, we selected instrumental SNP variables as the top 30 most significant SNPs after clumping all SNPs using a LD cutoff of 0.1.

## Analysis of WRAP data

WRAP is a longitudinal study of initially dementia-free middle-aged adults that allows for the enrollment of siblings and is enriched for a parental history of AD. Details of the study design and methods used have been previously described (*Johnson et al., 2018*; *Sager et al., 2005*). After quality control, a total of 1198 participants whose genetic ancestry was primarily of European descent were included in our analysis. On average, participants were 53.7 years of age (SD = 6.6) at baseline and had a bachelor's degree, and 69.8% (n=836) were female. Participants had two to six longitudinal study visits, with an average of 4.3 visits, leading to a total of 5184 observations available for analysis.

DNA samples were genotyped using the Illumina Multi-Ethnic Genotyping Array at the University of Wisconsin Biotechnology Center. Thirty-six blinded duplicate samples were used to calculate a concordance rate of 99.99%, and discordant genotypes were set to missing. Imputation was performed with the Michigan Imputation Server v1.0.3 (*Das et al., 2016*), using the Haplotype Reference Consortium (HRC) v. r1.1 2016 (*McCarthy et al., 2016*) as the reference panel and Eagle2 v2.3 (*Loh et al., 2016*) for phasing. Variants with a quality score $R^2$ <0.80, MAF <0.001, or that were out of HWE were excluded, leading to 10,499,994 imputed and genotyped variants for analyses. Data cleaning and file preparation were completed using PLINK v1.9 (*Chang et al., 2015*) and VCFtools v0.1.14 (*Danecek et al., 2011*). Coordinates are based on the hg19 genome build. Due to the sibling relationships present in the WRAP cohort, genetic ancestry was assessed and confirmed using Principal Components Analysis in Related Samples (PC-AiR), a method that makes robust inferences about population structure in the presence of relatedness (*Conomos et al., 2015*).

Composite scores were calculated for executive function, delayed recall, and learning based on a previous analysis (*Clark et al., 2016*). Each composite score was calculated from three neuropsychological tests, which were each converted to z-scores using baseline means and standard deviations. These z-scores were then averaged to derive executive function and delayed recall composite scores at each visit for each individual. Cognitive impairment status was determined based on a consensus review by a panel of dementia experts. Resulting cognitive statuses included cognitively normal, early MCI, clinical MCI, impairment that was not MCI, or dementia, as previously defined (*Koscik et al., 2016*). Participants were considered cognitively impaired if their worst consensus conference diagnosis was early MCI, clinical MCI, or dementia (n=387). Participants were considered cognitively stable if their consensus conference diagnosis was cognitively normal across all visits (n=803).

The 48 PRSs were developed within the WRAP cohort using PLINK v1.9 (*Chang et al., 2015*) and tested for associations with the three composite scores (i.e. executive function, delayed recall, and

learning) and cognitive impairment statuses. MCI status was tested using logistic regression models in R, while all other associations, which utilized multiple study visits, were tested using linear mixed regression models implemented in the lme4 package in R (*Bates et al., 2015*). All models included fixed effects for age and sex, and cognitive composite scores additionally included a fixed effect for practice effect (using visit number). Mixed models included random intercepts for within-subject correlations due to repeated measures and within-family correlations due to the enrollment of siblings.

## Software availability

The BADGERS software is freely available at https://github.com/qlu-lab/BADGERS, copy archived at *qlu-lab, 2024*.

## Acknowledgements

This project was supported by the Clinical and Translational Science Award (CTSA) program, through the NIH National Center for Advancing Translational Sciences (NCATS), grant UL1TR000427. Support for this research was also provided by the University of Wisconsin-Madison Office of the Chancellor and the Vice Chancellor for Research and Graduate Education with funding from the Wisconsin Alumni Research Foundation. BFD was supported by an NLM training grant to the Computation and Informatics in Biology and Medicine Training Program [NLM 5T15LM007359]. This research was also supported by the NIH [grants R01AG054047, R01AG27161, UL1TR000427, and P2C HD047873], Helen Bader Foundation, Northwestern Mutual Foundation, Extendicare Foundation, and the State of Wisconsin. The authors thank the University of Wisconsin Madison Biotechnology Center Gene Expression Center for providing Illumina Infinium genotyping services. We thank the International Genomics of Alzheimer's Project (IGAP) for providing summary results data for these analyses. The investigators within IGAP contributed to the design and implementation of IGAP and/or provided data but did not participate in analysis or writing of this report. IGAP was made possible by the generous participation of the subjects and their families. The i-Select chips were funded by the French National Foundation on Alzheimer's disease and related disorders. EADI was supported by the LABEX (laboratory of excellence program investment for the future) DISTALZ grant, Inserm, Institut Pasteur de Lille, Université de Lille 2, and the Lille University Hospital. GERAD was supported by the Medical Research Council (Grant n° 503480), Alzheimer's Research UK (Grant n° 503176), the Wellcome Trust (Grant n° 082604/2/07/Z), and German Federal Ministry of Education and Research (BMBF): Competence Network Dementia (CND) grant n° 01GI0102, 01GI0711, 01GI0420. CHARGE was partly supported by the NIH/NIA grant R01 AG033193 and the NIA AG081220 and AGES contract N01–AG–12100, the NHLBI grant R01 HL105756, the Icelandic Heart Association, and the Erasmus Medical Center and Erasmus University. ADGC was supported by the NIH/NIA grants: U01 AG032984, U24 AG021886, U01 AG016976, and the Alzheimer's Association grant ADGC–10–196,728. We thank contributors who collected samples used in this study, as well as patients and their families, whose help and participation made this work possible; Data for this study were prepared, archived, and distributed by the National Institute on Aging Alzheimer's Disease Data Storage Site (NIAGADS) at the University of Pennsylvania (U24-AG041689-01). We are also grateful for ADGC and its investigators for providing GWAS summary statistics for various AD phenotypes. The full acknowledgement to ADGC is included in the *Supplementary file 1*.

## Additional information

### Funding

| Funder | Grant reference number | Author |
| --- | --- | --- |
| National Center for Advancing Translational Sciences | Clinical and Translational Science Award (CTSA) program, UL1TR000427 | Bowen Hu Yunling Wang Qiongshi Lu Donghui Yan |

| Funder | Grant reference number | Author |
| --- | --- | --- |
| U.S. National Library of Medicine | Computation and Informatics in Biology and Medicine Training Program, NLM 5T15LM007359 | Burcu F Darst |
| National Institutes of Health | R01AG054047 | Corinne D Engelman Qiongshi Lu |
| National Institutes of Health | R01AG27161 | Sterling C Johnson |
| National Institutes of Health | UL1TR000427 | Burcu F Darst |
| National Institutes of Health | P2C HD047873 | Corinne D Engelman |
| NIH/NIA | U01 AG032984 | Alzheimer's Disease Genetics Consortium (ADGC) |
| NIH/NIA | U01 AG016976 | Alzheimer's Disease Genetics Consortium (ADGC) |
| NIH/NIA | U01 AG016976 | Alzheimer's Disease Genetics Consortium (ADGC) |
| Alzheimer's Association | ADGC–10–196,728 | Alzheimer's Disease Genetics Consortium (ADGC) |

The funders had no role in study design, data collection and interpretation, or the decision to submit the work for publication.

## Author contributions

Donghui Yan, Data curation, Software, Formal analysis, Validation, Visualization, Methodology, Writing – original draft, Writing – review and editing; Bowen Hu, Data curation, Formal analysis, Validation, Visualization, Methodology, Writing – original draft, Writing – review and editing; Burcu F Darst, Shubhabrata Mukherjee, Brian W Kunkle, Yuetiva Deming, Logan Dumitrescu, Adam Naj, Amanda Kuzma, Yi Zhao, Hyunseung Kang, Sterling C Johnson, Cruchaga Carlos, Timothy J Hohman, Paul K Crane, Corinne D Engelman, Data curation, Writing – review and editing; Yunling Wang, Data curation, Formal analysis, Writing – review and editing; Alzheimer's Disease Genetics Consortium (ADGC), Data curation; Qiongshi Lu, Conceptualization, Resources, Data curation, Formal analysis, Supervision, Funding acquisition, Validation, Investigation, Visualization, Methodology, Writing – original draft, Project administration, Writing – review and editing

## Author ORCIDs

Donghui Yan (ID) http://orcid.org/0000-0002-5433-0764
Yuetiva Deming (ID) http://orcid.org/0000-0001-7512-5703
Qiongshi Lu (ID) http://orcid.org/0000-0002-4514-0969

Reviewer #1 (Public Review): https://doi.org/10.7554/eLife.91360.2.sa1
Reviewer #2 (Public Review): https://doi.org/10.7554/eLife.91360.2.sa2
Author response https://doi.org/10.7554/eLife.91360.2.sa3

---

# Additional files

## Supplementary files

• Supplementary file 1. Simulation result; Result from Mendelian randomization and GSMR; Acknowledgements to Alzheimer's Disease Genetics Consortium (ADGC).

• Supplementary file 2. Full association result between UK-biobank traits and Alzheimer's disease/

endophenotypes.

- MDAR checklist

## Data availability

The current manuscript is a computational study, so no data have been generated for this manuscript. The modeling code is available at https://github.com/qlu-lab/BADGERS, copy archived at *qlu-lab, 2024*.

The following previously published datasets were used:

| Author(s) | Year | Dataset title | Dataset URL | Database and Identifier |
|-----------|------|---------------|-------------|-------------------------|
| Lambert JC | 2013 | NG00036 - IGAP Summary Statistics- Lambert et al. (2013) | https://www.niagads.org/datasets/ng00036 | NIAGADS database, ng00036 |
| Naj AC | 2011 | Alzheimer's Disease Genetics Consortium (ADGC) Collection | https://www.niagads.org/resources/related-projects/alzheimers-disease-genetics-consortium-adgc-collection | NIAGADS database, NG00027 |

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
