## [Editor Report · eLife assessment]

In the last 15 years, large-scale association studies (GWAS) have served to estimate the association between genome-wide common variants and a large number of disparate traits and diseases in humans. This **valuable** method provides a new way to find correlations between the genetic component of a phenotype of interest, and all this wealth of genetic information. This software adds as a new tool to investigate genetic correlation between traits, and to generate new mechanistic hypotheses and dissect the role of the observed associations in disease heterogeneity. The results of the application of their method are **solid** and generally agree with what others have seen using similar AD and UKB data.

---

## [Referee Report · Reviewer #1 (Public Review)]

The major aim of the paper was a method for determining genetic associations between two traits using common variants tested in genome-wide association studies. The work includes a software implementation and application of their approach. The results of the application of their method generally agree with what others have seen using similar AD and UKB data.

The paper has several distinct portions. The first is a method for testing genetic associations between two or more traits using genome-wide association tests statistics. The second is a python implementation of the method. The last portion is the results of their method using GWAS from AD and UK Biobank.

Regarding the method, it seems like it has similarities to LDSC, and it is not clear how it differs from LDSC or other similar methods. The implementation of the method used python 2.7 (or at least was reportedly tested using that version) that was retired in 2020. The implementation was committed between Wed Oct 3 15:21:49 2018 to Mon Jan 28 09:18:09 2019 using data that existed at the time so it was a bit surprising it used python 2.7 since it was initially going to be set for end-of-life in 2015. Anyway, trying to run the package resulted in unmet dependency errors, which I think are related to an internal package not getting installed. I would expect that published software could be installed using standard tooling for the language, and, ideally, software should have automated testing of key portions.

Regarding the main results, they find what has largely been shown by others using the same data or similar data, which add prima facie validity to the work The portions of the work dealing with AD subgroups, pathology, biomarkers, and cognitive traits of interest. I was puzzled why the authors suggested surprise regarding parental history and high cholesterol not associated with MCI or cognitive composite scores since the this would seem like the likely fallout of selection of the WRAP cohort. The discussion paragraph that started "What's more, environmental factors may play a big role in the identified associations." confused me. I think what the authors are referring to are how selection, especially in a biobank dataset, can induce correlations, which is not what I think of as an environmental effect.

Overall, the work has merit, but I am left without a clear impression of the improvement in the approach over similar methods. Likewise, the results are interesting, but similar findings are described with the data that was used in the study, which are over 5 years old at the time of this review.

---

## [Referee Report · Reviewer #2 (Public Review)]

Summary:

Yan, Hu, and colleagues introduce BADGERS, a new method for biobank-wide scanning to find associations between a phenotype of interest, and the genetic component of a battery of candidate phenotypes. Briefly, BADGERS capitalizes on publicly available weights of genetic variants for a myriad of traits to estimate polygenic risk scores for each trait, and then identify associations with the trait of interest. Of note, the method works using summary statistics for the trait of interest, which is especially beneficial for running in population-based cohorts that are not enriched for any particular phenotype (ie. with few actual cases of the phenotype of interest).

Here, they apply BADGERS on Alzheimer's disease (AD) as the trait of interest, and a battery of circa 2,000 phenotypes with publicly available precalculated genome-wide summary statistics from the UK Biobank. They run it on two AD cohorts, to discover at least 14 significant associations between AD and traits. These include expected associations with dementia, cognition (educational attainment), and socioeconomic status-related phenotypes. Through multivariate modelling, they distinguish between (1) clearly independent components associated with AD, from (2) by-product associations that are inflated in the original bivariate analysis. Analyses stratified according to APOE inclusion show that this region does not seem to play a role in the association of some of the identified phenotypes. Of note, they observe overlap but significant differences in the associations identified with BADGERS and other Mendelian randomization (MR), hinting at BADGERS being more powerful than classical top variant-based MR approaches. They then extend BADGERS to other AD-related phenotypes, which serves to refine the hypotheses about the underlying mechanisms accounting for the genetic correlation patterns originally identified for AD. Finally, they run BADGERS on a pre-clinical cohort with mild cognitive impairment. They observe important differences in the association patterns, suggesting that this preclinical phenotype (at least in this cohort) has a different genetic architecture than general AD.

Strengths:

BADGERS is an interesting new addition to a stream of attempts to "squeeze" biobank data beyond pure association studies for diagnosis. Increasingly available biobank cohorts do not usually focus on specific diseases. However, they tend to be data-rich, opening for deep explorations that can be useful to refine our knowledge of the latent factors that lead to diagnosis. Indeed, the possibility of running genetic correlation studies in specific sub-settings of interest (e.g. preclinical cohorts) is arguably the most interesting aspect of BADGERS. Classical methods like LDSC or two-sample MR capitalize on publicly available summary statistics from large cohorts, or having access to individual genotype data of large cohorts to ensure statistical power. Seemingly, BADGERS provides a balanced opportunity to dissect the correlation between traits of interest in settings with small sample size in which other methods do not work well.

Weaknesses:

However, the increased statistical power is just hinted, and for instance, they do not explore if LDSC would have identified these associations. Although I suspect that is the case, this evidence is important to ensure that the abovementioned balance is right. Finally, as discussed by the authors, the reliance on polygenic risk scoring necessarily undermines the causality evidence gained through BADGERS. In this sense, BADGERS provides an alternative to strict instrumental-variable based analysis, which can be particularly useful to generate new mechanistic hypotheses.

In summary, after 15 years of focus on diagnosis that would require having individual access to large patient cohorts, BADGERS can become an excellent tool to dig into trait heterogeneity, especially if it turns out to be more powerful than other available methodologies.

---

## [Author Response]

We thank eLife and the reviewers for the thoughtful summary and valuable review of our manuscript. We largely agree with the summary and review and have provided our responses to the comments below. We believe BADGER is a significant new tool for identifying associated risk factors for complex diseases, and the associations we observed in the analysis provide insights into the genetic basis of Alzheimer's disease.

**Reviewer #1 (Public Review):**
The major aim of the paper was a method for determining genetic associations between two traits using common variants tested in genome-wide association studies. The work includes a software implementation and application of their approach. The results of the application of their method generally agree with what others have seen using similar AD and UKB data.The paper has several distinct portions. The first is a method for testing genetic associations between two or more traits using genome-wide association tests statistics. The second is a python implementation of the method. The last portion is the results of their method using GWAS from AD and UK Biobank.

We thank the reviewer for the conclusion and positive comments.

Regarding the method, it seems like it has similarities to LDSC, and it is not clear how it differs from LDSC or other similar methods. The implementation of the method used python 2.7(or at least was reportedly tested using that version) that was retired in 2020. The implementation was committed between Wed Oct 3 15:21:49 2018 to Mon Jan 28 09:18:092019 using data that existed at the time so it was a bit surprising it used python 2.7 since it was initially going to be set for end-of-life in 2015. Anyway, trying to run the package resulted in unmet dependency errors, which I think are related to an internal package not getting installed. I would expect that published software could be installed using standard tooling for the language, and, ideally, software should have automated testing of key portions.

We thank the reviewer for their comments. To clarify, the primary difference between our proposed method, BADGERS, and LDSC lies in their respective objectives and applications. LDSC is designed to estimate heritability and genetic correlations between traits by utilizing GWAS summary statistics, thereby aiding in the elucidation of the genetic architecture of complex traits and diseases. Conversely, BADGERS is specifically developed to explore causal relationships between risk factors, such as biomarkers, and diseases of interest. It employs genetic variants as variables to deduce causality, thereby addressing the challenges of confounding and reverse causation that are common in observational studies. Although BADGERS utilizes the LD reference panel derived from LDSC, the LD reference panel is used to obtain the predicted trait expression. The ultimate goal is to focus on linking biobank traits with Alzheimer’s disease and building causal relationships instead of identifying genetic architecture.

Regarding the technical aspects mentioned, we acknowledge the concerns about the use of Python 2.7 and the issues encountered during the package installation. We are in the process of updating the software to ensure compatibility with current versions of Python and to enhance the installation process with standard tooling and automated testing for a more user-friendly experience. We have provided tests for each portion of the software so the user can test if the software is working properly.

Regarding the main results, they find what has largely been shown by others using the same data or similar data, which add prima facie validity to the work The portions of the work dealing with AD subgroups, pathology, biomarkers, and cognitive traits of interest. I was puzzled why the authors suggested surprise regarding parental history and high cholesterol not associated with MCI or cognitive composite scores since the this would seem like the likely fallout of selection of the WRAP cohort. The discussion paragraph that started "What's more, environmental factors may play a big role in the identified associations." confused me. I think what the authors are referring to are how selection, especially in a biobank dataset, can induce correlations, which is not what I think of as an environmental effect.

We thank the reviewer very much for their comment. We're glad that our findings align with existing research using similar data, increasing the validity of our work and the proposed BADGER algorithm. Your point about the lack of association between parental history, high cholesterol, and mild cognitive impairment (MCI) or cognitive composite scores in the WRAP cohort is well-taken. We agree that the selection criteria of the WRAP cohort may influence these findings, as it consists of individuals with a specific risk profile for Alzheimer's disease. This selection could indeed mitigate the observed association between these factors and cognitive outcomes, which we initially found surprising.

Regarding the environmental factors, we appreciate your clarification and understand the confusion. Our intention was to discuss the potential for selection bias and confounding factors in biobank datasets for the identified associations, which might not necessarily be direct environmental effects.

Overall, the work has merit, but I am left without a clear impression of the improvement in the approach over similar methods. Likewise, the results are interesting, but similar findings are described with the data that was used in the study, which are over 5 years old at the time of this review.

We thank the reviewer a lot for their endorsement of the BADGER framework. We believe that our method, BADGER, improves on existing approaches by effectively linking genetic data with the detailed phenotypic information in biobanks and large disease GWAS. This enhances our ability to detect associations without needing individual-level data, offering clearer insights while reducing issues like reverse causality and confounding factors.

Even though the IGAP dataset is over five years old, it remains one of the largest publicly available datasets for Alzheimer’s Disease. Likewise, the UK biobank is one of the largest publicly available human traits datasets, which researchers continue to use. These datasets' continued utility demonstrates their value in the research community. Additionally, the versatility of the BADGER framework makes it suitable for future research investigating the relationship between human traits and various diseases using different datasets.

**Reviewer #2 (Public Review):**
Summary:Yan, Hu, and colleagues introduce BADGERS, a new method for biobank-wide scanning to find associations between a phenotype of interest, and the genetic component of a battery of candidate phenotypes. Briefly, BADGERS capitalizes on publicly available weights of genetic variants for a myriad of traits to estimate polygenic risk scores for each trait, and then identify associations with the trait of interest. Of note, the method works using summary statistics for the trait of interest, which is especially beneficial for running in population-based cohorts that are not enriched for any particular phenotype (ie. with few actual cases of the phenotype of interest).Here, they apply BADGERS on Alzheimer's disease (AD) as the trait of interest, and a battery of circa 2,000 phenotypes with publicly available precalculated genome-wide summary statistics from the UK Biobank. They run it on two AD cohorts, to discover at least 14 significant associations between AD and traits. These include expected associations with dementia, cognition (educational attainment), and socioeconomic status-related phenotypes. Through multivariate modelling, they distinguish between (1) clearly independent components associated with AD, from (2) by-product associations that are inflated in the original bivariate analysis. Analyses stratified according to APOE inclusion show that this region does not seem to play a role in the association of some of the identified phenotypes. Of note, they observe overlap but significant differences in the associations identified with BADGERS and other Mendelian randomization (MR), hinting at BADGERS being more powerful than classical top variant-based MR approaches. They then extend BADGERS to other AD-related phenotypes, which serves to refine the hypotheses about the underlying mechanisms accounting for the genetic correlation patterns originally identified for AD. Finally, they run BADGERS on a pre-clinical cohort with mild cognitive impairment. They observe important differences in the association patterns, suggesting that this preclinical phenotype (at least in this cohort) has a different genetic architecture than general AD.

We thank the reviewer a lot for the conclusion and positive comments.

Strengths:BADGERS is an interesting new addition to a stream of attempts to "squeeze" biobank data beyond pure association studies for diagnosis. Increasingly available biobank cohorts do not usually focus on specific diseases. However, they tend to be data-rich, opening for deep explorations that can be useful to refine our knowledge of the latent factors that lead to diagnosis. Indeed, the possibility of running genetic correlation studies in specific sub-settings of interest (e.g. preclinical cohorts) is arguably the most interesting aspect of BADGERS. Classical methods like LDSC or two-sample MR capitalize on publicly available summary statistics from large cohorts, or having access to individual genotype data of large cohorts to ensure statistical power. Seemingly, BADGERS provides a balanced opportunity to dissect the correlation between traits of interest in settings with small sample size in which other methods do not work well.

We thank the reviewer a lot for the conclusion and positive comments.

Weaknesses:However, the increased statistical power is just hinted, and for instance, they do not explore if LDSC would have identified these associations. Although I suspect that is the case, this evidence is important to ensure that the abovementioned balance is right. Finally, as discussed by the authors, the reliance on polygenic risk scoring necessarily undermines the causality evidence gained through BADGERS. In this sense, BADGERS provides an alternative to strict instrumental-variable based analysis, which can be particularly useful to generate new mechanistic hypotheses.

We thank the reviewer a lot for the comments. We understand the importance of comparing BADGER to other methods. The comparison with LDSC, while not directly relevant toBADGER’s causal inference aims, is indeed an interesting aspect to consider for future studies. In this paper, we focused on comparing BADGER with Mendelian Randomization (MR), which shares its causal inference objective.

As a result, BADGERS identified a total of 48 traits that reached Bonferroni-corrected statistical significance. In contrast, MR-IVW only identified nine traits with Bonferroni-corrected statistical significance. Among these nine traits, seven were also identified by BADGERS. This demonstrates that BADGER holds higher power in detecting causal relationships.

Regarding the use of polygenic risk scoring, we agree that it holds challenges in directly inferring causality. While BADGERS offers an innovative way to explore genetic correlations and can help generate new hypotheses about disease mechanisms, it does not replace the causal inferences that can be drawn from instrumental-variable-based analyses. Instead, it should be viewed as a complementary tool that can illuminate potential genetic relationships and guide further causal investigations.

In summary, after 15 years of focus on diagnosis that would require having individual access to large patient cohorts, BADGERS can become an excellent tool to dig into trait heterogeneity, especially if it turns out to be more powerful than other available methodologies.

We thank the reviewer a lot for the conclusion and positive comments.